# ADVERSARY A3C FOR ROBUST REINFORCEMENT LEARNING

## ABSTRACT

Asynchronous Advantage Actor Critic (A3C) is an effective Reinforcement Learning (RL) algorithm for a wide range of tasks, such as Atari games and robot control. The agent learns policies and value function through trial-and-error interactions with the environment until converging to an optimal policy. Robustness and stability are critical in RL; however, neural network can be vulnerable to noise from unexpected sources and is not likely to withstand very slight disturbances. We note that agents generated from mild environment using A3C are not able to handle challenging environments. Learning from adversarial examples, we proposed an algorithm called Adversary Robust A3C (AR-A3C) to improve the agent's performance under noisy environments. In this algorithm, an adversarial agent is introduced to the learning process to make it more robust against adversarial disturbances, thereby making it more adaptive to noisy environments. Both simulations and real-world experiments are carried out to illustrate the stability of the proposed algorithm. The AR-A3C algorithm outperforms A3C in both clean and noisy environments.

## 1 INTRODUCTION

Robot control can become very difficult when the model is unknown or too complex; people tend to seek other ways to simplify the problem. A major way to simplify the complicated problem is to reduce the dimensionality using model generalization (Shuuji Kajita, 2005. Chaohui Gong, 2015) or linearize the nonlinear model in limited space (Cho, 2012. Luca, 1996). Another important method is reinforcement learning. Reinforcement learning is a set of ideas and methods that allow the robot agent to explore and acquire information from the environment to form a policy that gets the most cumulative reward from the environment. The environment gives feedback reward to represent the evaluation of actions that the robot agent takes. The agent updates itself by changing the probabilities over different actions under the same state to perform better in the next episode.

Reinforcement Learning algorithms (e.g., A3C) using Deep Neural Networks (DNNs) as function approximations have shown stunning performance in robot control (Mnih, 2016). However, the performances of those RL algorithms based on DNNs are not robust (Sandy Huang, 2017). Even with slight random disturbances, the policy tends to fail. We have observed a significant drop in cumulative reward when applying adversarial attack (see Figure 2). We aim to train a robust policy that is able to withstand an extremely baleful attack.

In this paper, we propose a method called Adversary Robust A3C (AR-A3C). Both simulation and real-world experiments show that AR-A3C has improved performance compared to the original A3C algorithm. We adopt a mechanism called Adversary Training (Morimoto and Doya, 2005). The idea is to strengthen the algorithm by adding an adversarial disturbance, which tries to fail the task, to the environment. The aim of this method is to make the robot agent get more experience over unexplored and harsh environment during the training process and thus react better in tests when disturbances present.

We test the AR-A3C algorithm through both simulation and real-world pendulum experiments (see Figure 1). To evaluate the robustness, we test the algorithm over different experiment settings, such as varying the mass and applying external blunt force. The policy learnt with adversarial training shows significant improvement in tests with and without adversarial disturbances.

The contributions of our work can be summarized as follows:

(a) We developed an algorithm (AR-A3C) that improves the robustness of the baseline A3C algorithm using adversary training.

(b) We built an pendulum hardware platform that enables automated adversarial training.

(c) We proved the robustness of AR-A3C through both simulation and real-world experiments. We also illustrated that model transfer of robust policy from simulation to physical model is feasible.

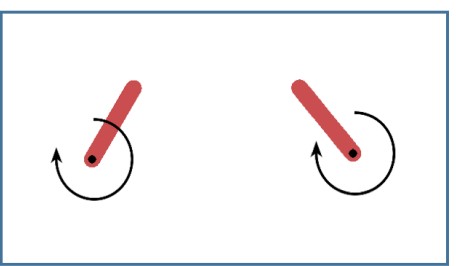 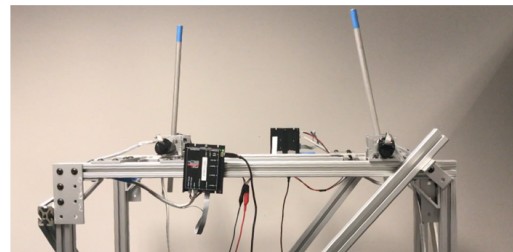

Figure 1: Dual pendulum experiments: simulation and hardware setup.

The remainder of this paper is organized as follows: In Section II, we introduce some background of our algorithm. Section III lists some related work in adversary agent field. Section IV explains our proposed AR-A3C method in details. In Section V, we evaluate our algorithm on both gym Mujoco tasks and real-world hardware. in Section VI, we conclude the present paper with future work and further topics.

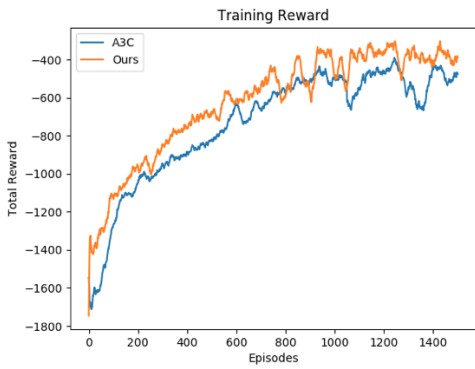 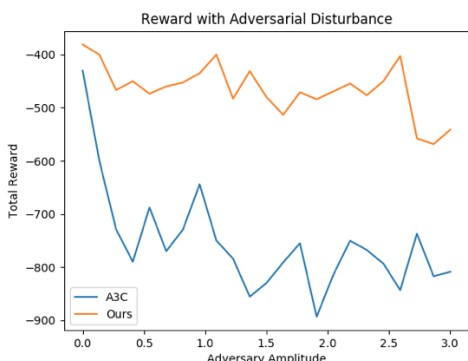

Figure 2: (a) Adversarial training to the same performance for A3C and AR-A3C in simulation, (b) A3C observes a significant drop in cumulative reward when applying adversarial attack while AR-A3C is not significantly affected by the adversarial attack.

## 2 PRELIMINARIES

Before diving deep into the AR-A3C, we first lay out some basic notions and terminology in reinforcement learning.

## 2.1 REINFORCEMENT LEARNING PROBLEM

The basic idea of reinforcement learning is to obtain an optimal policy $\pi_\theta^*$ that extracts as much cumulative reward $R$ as possible from the environment by choosing actions given a state. By policy, we mean a strategy that the decision maker (agent) follows to decide on actions based on parameterized rules given an input observation of the environment (state, $s$). The policy can be a set of weights that linearly combine the features in a state or neural networks. The environment in reinforcement learning context are well designed so that it provides the agent a new state $s'$ and reward $r$ after the agent takes a specific action. Each time step generates an experience tuple $(s, a, r, s')$. The policy optimize itself using the tuple batch$(s, a, r, s')_x$ every $x$ time steps, where $x$ is the predefined batch size. Normally, we need millions of time steps to reach an optimal policy $\pi_\theta^*$.

## 2.2 ASYNCHRONOUS ADVANTAGE ACTOR CRITIC (A3C)

A3C is a state-of-the-art algorithm for game and control problems (Mnih, 2016). It is derived from the Actor-Critic (AC) algorithm. Like AC, A3C works both in discrete and continuous action space environments. Actually, A3C is an asynchronous multi-thread version of actor critic algorithm.

AC combines both policy gradient and value gradient. The actor is a policy network that determines which action to be chosen at each time step. The critic contains a value network $V_\pi(s, a)$ that indicates how promising an action is under current state $s$. The critic outputs an evaluate value $V(s, a)$ for the actor, so that the actor will adjust its policy according to the evaluation. Both the actor and the critic will update their networks using the information provided by the environment. They get more and more accurate knowledge of the environment, based on which the agent's policy eventually converges to the optimal policy $\pi_\theta^*$.

A3C makes the AC algorithm easier and faster to converge. It creates multiple threads. Within each thread, an independent actor-critic pair interacts with the environment simultaneously. The unique exploration experience from each thread is sent to the global actor-critic pair. A3C eliminates the bias of a continuous experience trajectory by feeding only a small batch of experience tuple $(s, a, r, s')$ at one time, with several threads feeding at the same time.

## 2.3 ADVERSARY IDEA

The adversarial method is a way used in algorithm evaluation and algorithm strengthen. The solution to a zero-sum game is actually a minimax search; it is a result of Nash equilibrium. The players in the game try to minimize the maximum possible reward could be achieved by their opponents. Adversary training allows the algorithm meet more severe situations, especially those dangerous but unexplored situations, thereby making it more adaptable to noisy environments. In the zero-sum game setting, we call the agent that we wish to be more robust the protagonist, and the agent who tries to fail the task the adversary. Both the protagonist and the adversary receives feedback information from the environment and they both give an action based on the observation. The difference is that the adversary makes an action that tries to fail the task and make it get a lower reward.

## 2.4 POLICY TRANSFER

Training on hardware is not as easy as in simulation. The reason is that hardware operates in real-world; it must reset using some automatic or manual mechanism after each episode. It also takes lots of time because training a policy requires a huge amount of training data, and physics operates in the real-world is much slower than computer simulations nowadays. Besides, hardware can be easily broken if not treat correctly. Some tasks even require the robot to learn from crashing (Gandhi, et al. 2017), which is not likely to be the case in the real-world. This brings the problem of policy transfer from simulation to the real-world model. The problem is the fidelity of simulation, which always has some difference from the real model. Mechanisms that allow more accurate and efficient policy transfer have been proposed. (M. Cutler 2015, Clavera and Held 2017)

## 3    RELATED WORKS

Adversarial examples have been used in machine learning tasks like image classification and game playing. Christian Szegedy, et al. (2014) first found the adversarial example in the area of image classification that is able to fool the neural network to make a wrong decision. Nicolas Papernot, et al. (2016) posted a defensive distillation method, which reduces the effectiveness of adversarial sample creation to 0.5

Vahid Behzadan (2017) have shown the vulnerability of deep reinforcement learning algorithm by introducing the policy induction attacks. The neural network policy is easy to be fooled into a wrong action by introducing a little variation in the input. Sandy Huang (2017) also proved that the Adversarial Examples algorithm proposed in Goodfellow (2014) is able to affect reinforcement algorithms such as TRPO, A3C, and DQN. The tactic is to use uniform attack to affect the state at each timestep. Yen-Chen Lin (2017) proposed two different attack methods, named strategically-timed attack and enchanting attack respectively: (1) the strategically-timed attack reduces the frequency by setting a limit, so that the adversary attacks only if the expected reward exceeds the set threshold; (2) the enchanting attack tactic uses adversarial example method proposed by Carlini and wagner (2016) to fool the agent into a designated target state.

Adversary learning was firstly used as a structure to evaluate different algorithms. Littman (1994) proposed an adversary structure for Q-learning like algorithms to fit into a multi-agent game, thereby training the agents under Markov game and evaluating the performance of four different algorithms. William Uther (2003) proposed adversarial reinforcement learning which uses the same structure as in Littman (1994) to test then newly invented single-agent algorithm under the proposed two-player soccer game structure. Both of their adversary methods placed the agent in an equal opposite place; the aim of the structure is to compare the performance of different algorithms.

Another way to use adversary learning is to add robustness to agents. The zero-sum game was adopted by Morimoto and Doya (2005) in a method called Robust Reinforcement Learning (RRL). It introduced an adversary in the differential game based on the theory of $H_\infty$ control, the adversary tries to fail the game by minimize the reward through taking actions. The min-max solution takes the best policy and the adversary into account, thereby making the protagonist policy performs better (i.e., more robust) in the disturbance biased task.

Pinto (2017) extended the RRL using deep RL method (such as TRPO) as a function approximation. The method, called Robust Adversary Reinforcement Learning (RARL), adopts the adversary training by introducing the same adversary agent as the protagonist in the training process. RARL improved the robustness of policy based algorithm for robotics tasks.

## 4    ADVERSARY ROBUST A3C

### 4.1    ADVERSARIAL TRAINING

The task we are formulating is a zero-sum dual-agent markov game. Markov means a feature that the agent can determine which action to choose based only on the current state, without the necessary to care about previous states. This feature is common for most RL environments. Zero-sum game provides a mechanism for two agents to compete against each other to achieve their respective goals. For example, one agent aims to get the most cumulative reward from the environment, while the other tries to fail the task by reducing the reward. The mechanism in zero-sum game results in a minimax solution. How can we find the minimax solution under the game setting? We can initialize the policy networks for these two agents and train them under the game setting simultaneously. These two agents, protagonist and adversary, compete against each other to achieve opposite goals. With protagonist always taking the best control output and adversary always making the worst disturbing output, the game requires the protagonist to achieve the same performance in the environment with adversary.

Pinto (2017) noted that adversary method such as changing friction or mass can be represented as adversarial forces. We treat the output of adversary policy as an external force and apply it to the model. We also observed that with different levels of adversarial magnitude, the reactions of the training result are different. The game is considered to be hard for the protagonist if the

adversarial magnitude is high. If the magnitude is low, the environment becomes mild as if there is no troublemaking adversary. We tune the magnitude to a degree that the protagonist receives enough adversarial training while still be able to achieve the same performance. Another challenge is how does the adversary pick an action that generates the worst result. Instead of using the pre-settled model-based policy, we make the adversary also trainable. This is achieved by using the feature of zero-sum game. We feed the negative reward into the adversarial agent so that the agent updates itself trying to maximize the negative reward, which is actually reducing the reward for the protagonist. One benefit of the trained policy is that it does not make any priori knowledge of the environment; it learns to pick the worst action directly from the reward feedback, which is based on the system transition dynamics.

In the A3C setting, each thread has an actor-critic pair. While in AR-A3C, we extend each thread from a single agent to double agents, one protagonist and one adversary. The adversary can be other learning algorithms besides Actor Critic. But here we choose Actor Critic as our adversary, so that the adversary and the protagonist have the same network structure. It is easy to implement and the effect is promising because there is already a good enough policy for the adversary to adopt: one can always generate a contradictory force against the protagonist. The adversary should learn a policy at least better than the inversed force.

## 4.2 FORMULA

The Protagonist policy $\pi_\mu$ and Adversary policy $\pi_\nu$ both participate in the the zero-sum game. We also introduce the difficulty level parameter $D$ which confines the magnitude of adversarial output. The actions of the protagonist and the adversary both generate impact on the environment and affect the next state $s'$ according to the system dynamics $P$.

$s' = P(s, a, a', u)$, where $u$ is the system inherent noise, $s'$ is the new state. During the rollout process, each time step forms a tuple $(s, a, a', r, s')$. The trajectory of the experience can be represented in batches of tuples. we feed batches into the protagonist and adversary agents to update their policies. The expected reward $R(s, a)$ is the cumulative discounted reward.

$$R(s, a) = \mathbb{E}(\sum_{t=0}^{T}(y^t * r(s_t, a_t))) = \mathbb{E}(r + y * V(s')).$$

We also have the estimated reward $V(s)$. We define the advantage $A(s, a)$ as the difference between the expected reward and estimated reward: $A(s, a) = R(s, a) - V(s)$. In zero-sum games, the total reward for all agents is zero. So we have $r_{protagonist} = -r_{adversary} = r$.

We then update the protagonist policy $\mu$ and the adversary policy $\nu$ as follows.

$$\theta_\mu = \theta_\mu + \alpha * A_{protagonist} * \nabla_{\theta_\mu} log(\pi_\mu); \theta_\nu = \theta_\nu + \alpha * A_{adversary} * \nabla_{\theta_\nu} log(\pi_\nu).$$

## 4.3 SOFTWARE PART

We choose the adversary to have the same-actor critic pair structure as the protagonist does. Both the protagonist and the adversary hold the same NN structure: 1-layer 100-nodes for the critic and 1-layer 200-nodes for the actor. The adversary outputs an action based on the observed information, which aims to minimize the cumulative reward. So we feed the negative of the reward so that the AC pair will try to maximize the minus reward by applying gradient update to its network.

(Mnih, 2016) showed that the number of threads would not significantly affect the final reward. In the training process, we use two threads for A3C, in order to be consistent with the hardware setup. Within each thread, we have one protagonist AC pair and one adversary AC pair. They both apply actions on the model in the same time step. We would like to address the difference between the adversarial training with RARL (Pinto et al., 2017). For one batch, we update both the protagonist and the adversary AC pairs, which can reduce the rollout episodes by half.

We use Tensorflow to build the network and use threading in Python to create multiple threads for AR-A3C. In simulation, we use the Mujoco (Todorov et al., 2012) pendulum model to test our algorithm. The pendulum is a continuous action space task, with three features in the state and 1-DoF continuous action output. The simulated pendulum model enables automatic continuous

training, which does not require human intervention. It should be pointed out that we also build a hardware dual pendulum hardware platform capable of asynchronous automatic training.

Each thread interacts with the same environment, conducted by a worker. At each time step, the environment send the current state information to the worker. The worker collect the actions from the protagonist and the adversary, send them back to environment and wait for the new state update. The worker pushes its experience after collect a batch of time steps or at the end of the episode. The global network updates its policy and value networks, then transfers the new weights to the worker. Each thread works simultaneously, until the global policy converges to the optimum. The communication mechanism is shown in Figure 3.

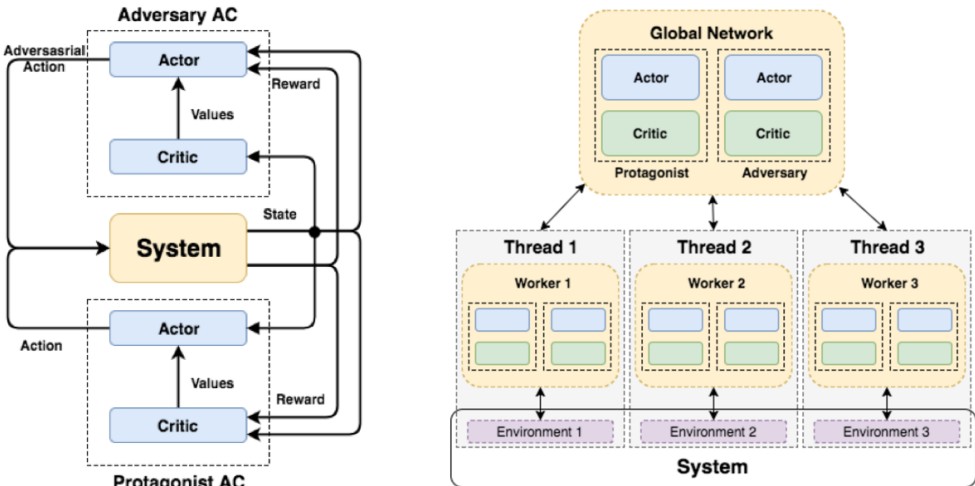

Figure 3: (a) A single actor-critic mechanism. The system is the environment. The protagonist and the adversary apply their actions to the system and they will get the same state observation and the same reward from the system, based on which they take the next move. (b) AR-A3C Architecture. Each thread contains a protagonist AC pair and an adversary AC pair. They both interact with the system environment and update the global network using their own experiences.

## 4.4 HARDWARE PART

To evaluate policy transfer from simulation to the real-world model, we build a dual pendulum platform. As shown in Figure 4, the platform consists of the two pendulums that can freely rotate without interfering with each other. For each pendulum module, there are two Maxon DC motors aligned along the same rotational axis: one motor acts as the protagonist and the other one serves as the adversary. The benefit of using two motors is that both the protagonist and adversarial forces can be applied to the pendulum conveniently. In addition, the double-supported pendulum could avoid the wiggling and instability issues introduced by the cantilever beam form, especially for high-speed operations.

When training the A3C algorithm, the adversary motor is connected to the pendulum without providing any active rotational torque (using zero current/torque control), ensuring that A3C has the same friction as AR-A3C does during training. A major difference between AR-A3C and A3C is the presence of external adversarial force.

The pendulum is a 500mm long aluminum rod. It weighs 271g, and its moment of inertia is $5.659 * 10^{06} g * mm^2$. The output of the protagonist agent is a number ranged between [-2, 2], which is equivalent to [-3000, 3000]mA variation on the motor current. The maximum torque generated by the maxon motor is not able to swing up the pendulum directly from its rest location. The agent needs to swing at least three times to inject enough energy to pendulum to swing it up.

---

**Algorithm 1** pseudocode for AR-A3C(Proposed Algorithm)

---

1.
1: **Initialize**: Global Episodes Number $N$, BatchSize $x$
2: **Initialize**: n Workers $W_i$                                                     $\triangleright i \in [1, n]$
3: **Initialize**: Protagonist Agent Policy $\mu_i$ and Adversary Agent Policy $\nu_i$ for $W_i$
   $\triangleright \mu_i$ and $\nu_i$ are parameterized neural network
4: **Initialize**: Global Agent Policy $\mu_g$ and $\nu_g$
5: **for** $i = 1$ to $n$ **do**
6:     $W_i.startWorkingThread()$
7: **end for**
8: **function** STARTWORKINGTHREAD
9:     **for** $j = 1$ to $N$ **do**
10:       $[s, a_\mu, a_\nu, r, a']_x = $ Rollout$(E_i, \mu_i, \nu_i)$
11:       $\mu_g = $ policyOptimizer$(s, a_\mu, r, s')$                 $\triangleright$ Here we used RMSPropOptimizer
12:       $\nu_g = $ policyOptimizer$(s, a_\nu, r, s')$
13:       $\mu_i = \mu_g$
14:       $\nu_i = \nu_g$
15:     **end for**
16:     **Return**: $\mu_g, \nu_g$
17: **end function**
18: **function** ROLLOUT$(E_i, \mu_i, \nu_i)$
19:     $s = E_i.reset()$
20:     **for** $k = 1$ to $x$ **do**
21:       $[a_\mu, a_\nu] = [\mu_i.chooseAction(s), \nu_i.chooseAction(s)]$
22:       $[s', r] = E_i([a_\mu + a_\nu])$                 $\triangleright$ Add Adversarial Force to the model
23:       $trajectory.append([s, a_\mu, a_\nu, r, a'])$
24:       $s = s'$
25:     **end for**
26:     **Return**: $trajectory$
27: **end function**

---

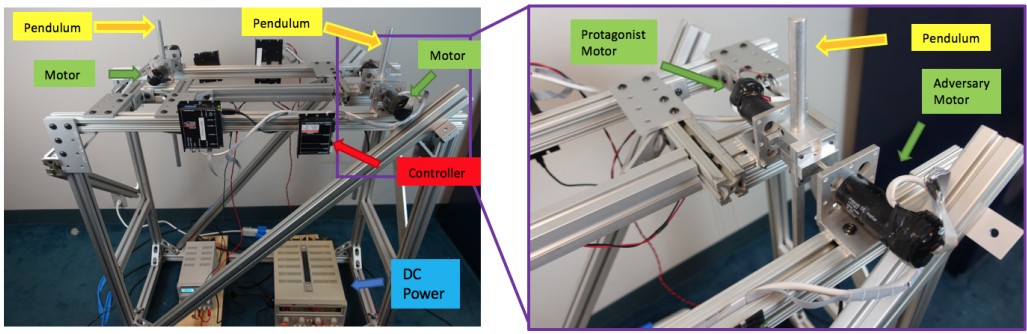

Figure 4: The protagonist and adversary motors in the hardware setup.

Since directly training the policy on hardware is hard, we decide not to train the policy in real-world from scratch. The method we adopted uses further training to ensure the policy converge to optimum in the new model. The learnt policy in simulation does not perfectly fit the real model, but it does increase the learning efficiency by providing intelligent prior information. we transfer our learnt model from simulation to real world by further training for several episodes. this allows the policy to tune its parameter to fit the new model and achieve the same performance as in the simulation. Based on a good initialization from the simulator, the policy is able to converge quickly to the optimum.

# 5 EXPERIMENT

The robustness of the proposed method is evaluated through both simulation and real-world hardware experiments.

## 5.1 ENVIRONMENT SETTINGS

We wrap the maxon motor controller into an environment using ROS (Robot Operating System). The agent gets reward from the environment based on the current state, which includes $cos(\theta)$, $sin(\theta)$ of the current angle and the angular velocity $d_\theta$.

$$s = [cos(\theta)sin(\theta)d_\theta].$$

The reward indicates whether the current state-action pair is good or not, by combining the information in $s$.

$$r = -(\theta^2 + 0.1 * d_\theta^2 + 0.001 * a^2)$$

, where $a$ is the action for the current step. Due to the soft limit to the position and velocity, the reward is a number ranging over [-16.27, 0]. Better state-action pair tend to give higher rewards.

## 5.2 INITIALIZATION

To make the agent experience more states, we randomly pick the initial state rather than always starting from the same position and velocity. This allows the agent to experience more situations, thereby eliminating the bias of the repetition in the initial state.

## 5.3 DEBUGGING ISSUES

We have observed that after training 300 episodes from scratch, the pendulum starts to rotate like crazy. So we set limits on the rotational speed to prevent the pendulum from broken. The DC motor tends to get super hot after 30-minute continuous operation. To mitigate this problem, we put cooling pack on the motor, preventing it from overheating.

The agent communicates with the environment at a 30-Hz frequency, writing higher level command to the controller. The motor is operated at torque control mode. The Maxon EPOS2 control card is actually performing lower-layer control of the motor current, which is supposed to be proportional to the output torque in theory. The model-free RL algorithm should be able to handle the slight nonlinearity.

## 5.4 ROBUSTNESS

We have performed several set of experiments in simulation and on hardware to illustrate the robustness.

In Figure 2(b), we apply adversarial force to the trained policy, for both A3C and AR-A3C, in simulation. The Y axis is the reward gained under the adversary's impact. The X axis is the magnitude of the adversary, which indicate how big the impact is. We observe a huge drop in the reward for A3C, even with a very light adversary attack. In comparison, AR-A3C is more robust against the attack.

Moreover, we vary the weight and the moment of inertia by adding a clog onto the end of the real pendulum, as shown in Figure 6. We can also adjust the weight of the clog (20g, 40g, 60g, 80g). As is shown in the Figure 5, when x = 0, i.e., the environment has not changed the weight yet, our algorithm has a similar performance to A3C. As the weight of clog increases, A3C quickly dysfunctions while AR-A3C gets much higher reward on average. Therefore, AR-A3C is more tolerant to parameter variations on the weight and inertia of the pendulum.

In Figure 7, we apply external impact force using the adversary motor and record the reward data in the first 1000 time step. The reward can best indicate the current position and state. Zero reward means the pendulum is heading upwards while negative reward mean is pendulum is struggling waving up. From Figure 7, we see that after external impact, AR-A3C is able to quickly recover while A3C took a long time spinning before it returns upwards.

## 5.5 Hardware Training

The training policy is started from a 1500 episodes in simulation. We transfer the policy on to our hardware platform. The input and output of the policy is the same. With the priori knowledge, the policy on hardware quickly converges after 300 episodes on the hardware.

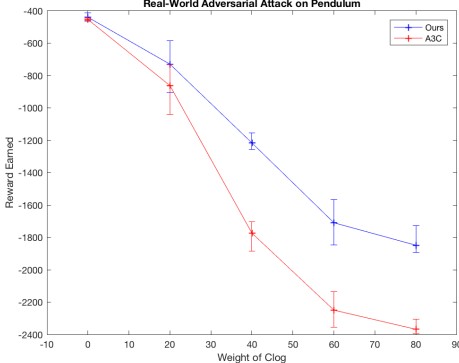

Figure 5: Evaluation of the robustness by varying the attached clog weights to the real pendulum hardware.

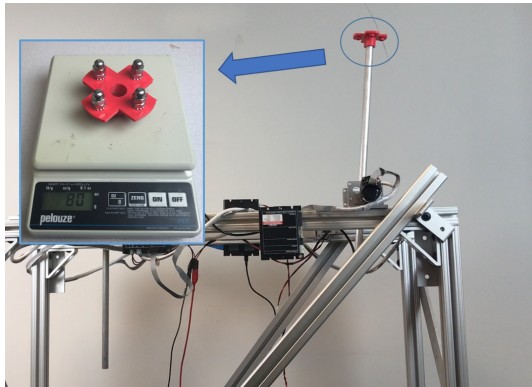

Figure 6: A clog is attached to the end of the pendulum to vary its weight and rotational inertia. The electronic scale shows the weight of the clog.

## 5.6 Visualization

We plot the model to visualize how the adversary works by showing the pendulum's velocity and the actions adopted by agents, as shown in Figure 8.

We observe that the adversarial force is not always opposite to the protagonist force. It is intuitively correct because pushing may generate more destruction power than pulling at certain states. For example, Figure 8(d) indicates that in order to fail the task, the adversarial force should generate a force that has the same direction as the protagonist does. It seems that the adversary is helping the pendulum to get to the top position, but actually the adversary is trying to accelerate the pendulum so that is will pass the top and fall again. The adversary is smart to utilize the model dynamics to fail the task.

## 5.7 Hardware Video

A companion video that compares the baseline A3C and our proposed AR-A3C algorithms is included in the following link. `https://youtu.be/95D-V9ybPmI`

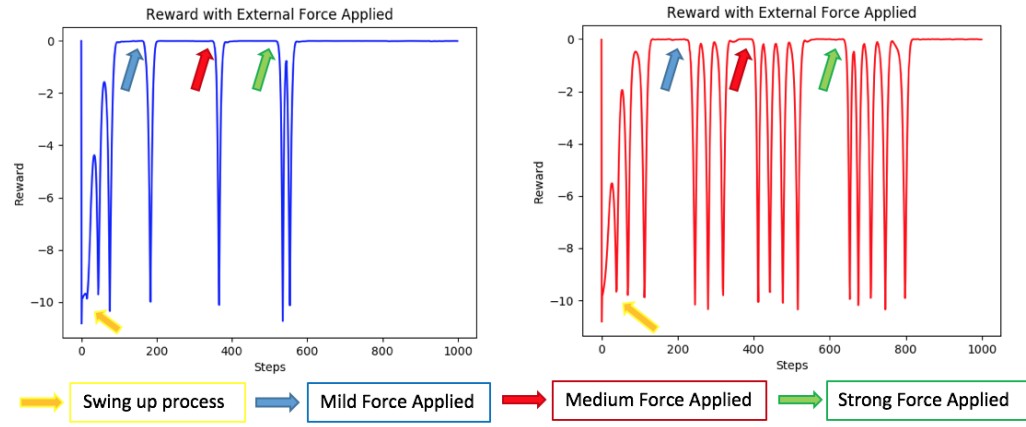

Figure 7: Rewards over different time steps. (a) External impact force applied to AR-A3C. (b) External impact force applied to A3C.

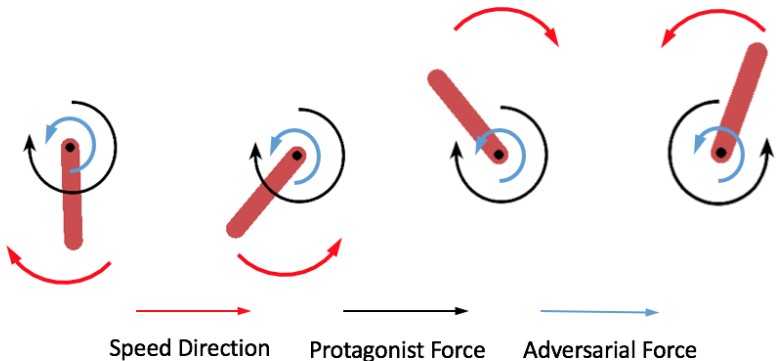

Figure 8: Visualization of the model with adversarial force and velocity.

## 6 CONCLUSIONS AND FUTURE WORK

In this paper, we developed an algorithm (AR-A3C) that improves the robustness of the baseline A3C algorithm using adversary training. We built an pendulum hardware platform that enables automated adversarial training. We also adopted a method that transfer the model from simulation to hardware from efficient learning. We proved the robustness of AR-A3C through both simulation and real-world experiments. It should be noted that the hardware platform developed in this paper can also be used to test other algorithms because it has implemented continuous automatic training.

Further work includes a multi-level training, which trains the model in an environment where the noise level gradually increases. This framework allows the policy to challenge itself to quantify the degree of robustness it can achieve. We can also tune the hardness level by changing the maximum external force that can be applied onto the model to make the algorithm find its most robust policy.

ACKNOWLEDGMENTS

The authors express their thanks to Mujoco and TensorFlow for the support of non-profit academic research.

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
