# OpenReview forum: "Adversary A3C for Robust Reinforcement Learning"
_ICLR.cc/2018/Conference — Reject_

### Official Review · AnonReviewer2 · 2017-11-27
**Interesting approach and hardware validation, but methods and comparisons are lacking**

**Rating:** 4
**Confidence:** 4

**Review:**

Positive:
- Interesting approach
- Hardware validation (the RL field needs more of this!)

Negative:
- Figure 2: what is the reward here? The one from Section 5.1?
- No comparisons to other methods: Single pendulum swing-up is a very easy task that has been solved with various methods (mostly in a cart-pole setup). Please compare to existing methods such as PILCO, basic Q-learning, classical methods...
- I'm not sure what's going on with the grammar in Section 5.3 ("like crazy", "super hot"...). This section also seems irrelevant (move to an appendix/supplementary or remove).
- You should plot a typical control curve for the motors (requested torques). This might explain your heat problem (I'm guessing the motor is effectively controlled by a bang-bang controller).
- Why did you pick this task? It's fine to only validate on a single task in hardware, but why not include additional simulation results (e.g. double pendulum)?

---

### Official Review · AnonReviewer1 · 2017-11-28
**The proposed technique is of modest contribution and the experimental results do not provide sufficient validation of the approach.**

**Rating:** 4
**Confidence:** 4

**Review:**

The authors propose an extension of adversarial reinforcement learning to A3C. The proposed technique is of modest contribution and the experimental results do not provide sufficient validation of the approach.

The authors propose extending A3C to produce more robust policies by training a zero-sum game with two agents: a protagonist and an antagonist. The protagonist is attempting to achieve the given task while the antagonist's goal is for the task to fail.

The contribution of this work, AR-A3C, is extending adversarial reinforcement learning, namely robust RL (RRL) and robust adversarial RL (RARL), to A3C. In the context of this prior work the novelty is extending the family of adversarial RL methods. However, the proposed method is still within the same family methods as demonstrated by RARL.

The authors state that AR-A3C requires half as many rollouts as compared to RARL. However, no empirical comparison between the two methods is performed. The paper only performs analysis against the A3C and no other adversarial baseline and on only one environment: cartpole.  While they show transfer to the real world cartpole with this technique, there is not sufficient analysis to satisfactorily demonstrate the benefits of the proposed technique.

The paper reads well. There are a few notational issues in the paper that should be addressed. The authors mislabel the value function V as the  action value, or Q function. The action value function is action dependent where the value function is not.  As a much more minor issue, the authors introduce y as the discount factor, which deviates from the standard notation of \gamma without any obvious reason to do so.

Double blind was likely compromised with the youtube video, which was linked to a real name account instead of an anonymous account.

Overall, the proposed technique is of modest contribution and the experimental results do not provide sufficient validation of the approach.

---

### Official Review · AnonReviewer5 · 2017-12-18
**Ok but not good enough**

**Rating:** 4
**Confidence:** 4

**Review:**

Clarity
The paper is clear in general.

Originality
The novelty of the method is limited. The proposed method is a simple extension of L. Pinto et al. by replacing TRPO with A3C. No evidence is provided to show the proposed method is competitive with the original TRPO version.

Significance
- The empirical results on the hardware are valuable.
- The simulated results are very limited. The neural networks used in the simulation have only one hidden layer. The method is tested on the Pendulum domain.

Pros:
- Real hardware results are provided.

Cons:
- Limited simulation results.
- Lacking technical novelty.

---

### Decision · Program_Chairs · 2018-01-29
**ICLR 2018 Conference Acceptance Decision**

**Decision:**

Reject

**Comment:**

Reviewers are unanimous in scoring this paper below threshold for acceptance.  The authors did not submit any rebuttals of the reviews.

Pros:
Paper is generally clear.
Hardware results are valuable.

Cons:
Limited simulation results.
Proposed method is not really novel.
Insufficient empirical validation of the approach.